# VEFLOW: TRAINING-FREE TEXT TO VIDEO EDITING VIA INVERSION-FREE VIDEO EDITING FLOW

## ABSTRACT

We present VEFLOW, a novel flow-based training-free text-driven video editing framework, which enables editing the video content based on the editing prompt in an inversion-free manner. Departing from existing training-free methods that rely on an "invert-then-edit" pipeline, we build upon the flow-based generative model and derive a novel video editing flow governed by a dedicated ordinary differential equation (ODE), which transforms the source video into the target video directly within the data space, thereby eliminating the need for explicit inversion. This new paradigm enables precise control over the editing region and inspires us to develop the attention-guided flow masking (AFM) module to suppress the unintended alteration. It masks out the undesired editing flow by identifying the region where the edit occurs via the cross-attention mask extracted from the source and target editing flow estimation. Besides, we observe that the estimated video editing flow may also lead to insufficient editing results due to the conflicts between the source and target flows. To tackle this issue, we further design the decoupled flow modulation (DFM) module to mitigate the potential editing conflict and enhance the editing performance through flow projection and modulation. By incorporating these designs, our approach demonstrates significant superiority over existing methods, particularly in editing efficiency, background preservation, and content editability. Extensive experiments on real-world videos confirm the effectiveness of our approach, offering a fresh perspective on text-driven video editing.

## 1 INTRODUCTION

With the advancement of generative models, particularly driven by diffusion models Ho et al. (2020); Song et al. (2021); Podell et al. (2023); Chen et al. (2023b), text-to-video (T2V) generation has seen remarkable progress with numerous advanced T2V models Guo et al. (2024); Blattmann et al. (2023); Yang et al. (2024); Kong et al. emerging, demonstrating the impressive capability to produce high-quality, temporally coherent video clips. As a crucial bridge between T2V generation and practical content creation, text-driven video editing aimed at modifying video content based on user-provided textual prompts has garnered increasing attention.

Following the development of text-to-video generation, a variety of text-driven video editing methods have also emerged, which can be broadly classified into training-free Qi et al. (2023); Geyer et al. (2023); Kara et al. (2024); Cong et al. (2023); Wang et al. (2025), one-shot tuning Wu et al. (2023), and learning-based Liu et al. (2024) approaches. This paper primarily focuses on the training-free approach. Inspired by the success of the "invert-then-edit" pipeline in text-driven image editing Mokady et al. (2023); Hertz et al. (2022); Avrahami et al. (2025), most training-free video editing methods adopt a similar routine: first, inverting the source video into a noise latent representation via inversion techniques like DDIM Inversion Song et al. (2020), and then performing text-guided denoising to achieve the desired edits. The underlying principle is that the inverted latent noise captures the essential features of the source video, allowing for modifications guided by the editing prompt while preserving as much of the original content as possible. Early efforts, such as TokenFlow Geyer et al. (2023) and FLATTEN Cong et al. (2023), rely on text-to-image (T2I) to achieve video inversion and primarily focus on maintaining temporal consistency throughout the editing process. More recently, VideoDirector Wang et al. (2025) points out that the temporal inconsistency stems from inaccurate T2I-based video inversion and proposes to achieve more accurate video inversion with the T2V model through the spatio-temporally decoupled guidance.

Despite their impressive achievements, inversion-based video editing methods suffer from two primary limitations: (1) Unintended content alteration. While recent works have made considerable efforts to enable temporally consistent video editing, they still struggle to maintain the unedited region consistent with the source video, especially when handling complex scenes or complicated motion. On one hand, it is caused by the inherent reconstruction error from the video inversion. On the other hand, it also lacks an effective scheme to make a precise edit region control within the inversion-based framework. (2) Inferior editing efficiency. These inversion-based approaches typically require first inferring an ideal latent noise through inversion before performing edits. This sequential process hinders parallelization and slows down the editing. The problem is more prominent for the method that employs additional optimization techniques Mokady et al. (2023) during the inversion process. For instance, VideoDirector Wang et al. (2025) optimizes multi-frame multi-text embeddings to minimize video inversion errors. Although effective, it requires iterative gradient updates at each inversion step, which significantly impairs editing efficiency.

In this work, we propose VEFLOW, a novel inversion-free video editing framework that addresses the key limitations of existing methods. Inspired by recent advances in flow-based generative models Esser et al. (2024); Liu et al. (2022); Kulikov et al. (2024), we model the source-to-noise and target-from-noise transformation as the flow procedure. Instead of separating these two flow paths, we conceptualize the editing process by compositing these two flows, which directly models the transition from source to target. This is achieved by deriving a dedicated video editing flow governed by an ordinary differential equation (ODE), which estimates the desired edit at each step from source video to target video within the data space. This new editing paradigm endows us with superior flexibility to precisely control the editing region as *the desired edit we estimated is directly applied to the source video instead of noise latent.* Build upon this feature, we develop the attention-guided flow masking (AFM) module to suppress the unintended content change. Specifically, AFM estimates the editing region from the cross-attention map produced during source and target flow estimation, including the region of the source object occupied and the target object that is going to emerge. The obtained mask is then utilized to mask out the editing flow of the unedited regions, thereby suppressing the alterations in unintended areas, such as the background. Besides, we observe that the insufficient video editing results may also occur when editing with the composited editing flow in some cases. We attribute this to the potential conflicts between the estimated source and target flow due to their distinct behavior (source disappearance and target emergence). To tackle this issue, we design the decoupled flow modulation (DFM). Specifically, we project the target flow toward the source flow to obtain the component conflict with the desired target transformation, and then, we mitigate such a component from the target flow to obtain a more concentrated editing direction. Afterwards, we strengthen the obtained editing flow with a coefficient to enhance the target editing force. Such flow projection and modulation drive the source video to transform toward the target more effectively. By incorporating these designs, our approach offers a simple but effective video editing solution, with exceptional performance compared to the existing methods in terms of editing efficiency, background preservation, and content editability. Extensive experiments with the real-world videos demonstrate the effectiveness of our approach. Overall, our contributions are summarized as:

- We introduce a novel flow-based inversion-free video editing framework, which eliminates the explicit inversion process, offering a new perspective on text-driven video editing.

- We develop two effective modules based on this new paradigm, including attention-guided flow masking and decoupled flow modulation, to achieve background preserving and effective video editing.

- Extensive experiments demonstrate the effectiveness of our method, significantly outperforming state-of-the-art methods across various real-world video editing scenarios.

## 2 METHODOLOGY

### 2.1 PRELIMINARY

**Flow-based Generative Models**. The generative flow models attempt to construct a transportation path between the distributions of two random vectors, $X_0$ and $X_1$. Generally, this process can be described via an ordinary differential equation (ODE) over time $t \in [0, 1]$,

$$dZ_t = V(Z_t, t)dt. \tag{1}$$

Here, $V$ is a time-dependent velocity field, parameterized by our flow model $v_\theta$. Usually, we set $X_0$ to the data distribution and $X_1$ to $N(0, I)$. The Eq. (1) transforms the sample of $X_1$ into the sample from $X_0$. This is done by initializing the ODE at $t = 1$ with a standard Gaussian sample, and numerically solving the ODE backwards in time with the velocity estimated by $v_\theta$ until reaching time $t = 0$ to obtain a sample $Z_0$ which follows the distribution of $X_0$. Rectified flow models Liu et al. (2022); Esser et al. (2024) are a particular type of flow models. It constructs a special ODE path such that the marginal at time $t$ corresponds to a linear interpolation between $X_0$ and $X_1$, *i.e.*, $Z_t = (1 - t)X_0 + tX_1$. In modern flow-based image and video generative models, we consider the conditional flow model $v_\theta(z_t, t, c)$, where $z_t$ is the sample along the flow path, $c$ is the prompt that we utilize to specify the content we intend to generate. Refer to Sec. A.2 for more explanation.

## 2.2 VEFLOW- INVERSION-FREE VIDEO EDITING FLOW

Given the source video $X^{\text{src}}$ with corresponding prompt $c^{\text{src}}$, we aim to modify the video content based on the target editing prompt $c^{\text{tar}}$. Instead of take the inversion-based approach, we build upon the flow-based video generative model and derive the inversion-free video editing flow to achieve more efficient video editing.

**Direct Video Editing Flow**. Inspired by Kulikov et al. (2024), we want to construct a flow path directly from the source video to the target edited state, rather than traversing through a noise space. Formally, our objective is to find a editing flow path $Z_t^{\text{edit}}$ from $X^{\text{src}}$ to $X^{\text{tar}}$ with $Z_1^{\text{edit}} = X^{\text{src}}$ and $Z_0^{\text{edit}} = X^{\text{tar}}$. Let's firstly consider the source and target flow ODEs,

$$dZ_t^{\text{src}} = V(Z_t^{\text{src}}, t)dt, \quad dZ_t^{\text{tar}} = V(Z_t^{\text{tar}}, t)dt, \tag{2}$$

where $Z_0^{\text{src}} = X^{\text{src}}$, $Z_0^{\text{tar}} = X^{\text{tar}}$ and $Z_1^{\text{src}} = Z_1^{\text{tar}} = N(0, I)$. These two paths describe the conversion between the source/target distribution and the standard Gaussian distribution. As shown in Fig. 1, the common video editing methods first invert the source video into the Gaussian noise that encodes the source input information, and then denoise toward the target video with the guidance of editing prompt, *i.e.*, $Z_0^{\text{src}} \xrightarrow{c_{\text{src}}} Z_1^{\text{src}} = Z_1^{\text{tar}} \xrightarrow{c_{\text{tar}}} Z_0^{\text{tar}}$. However, inverted noise latent not only introduces the reconstruction error, but also makes the precise editing region control infeasible. To this end, we construct a special editing flow path based on Eq. (2) as follows:

$$Z_t^{\text{edit}} = Z_0^{\text{src}} + Z_t^{\text{tar}} - Z_t^{\text{src}}. \tag{3}$$

Apparently, such edit flow path satisfy our editing boundary condition $Z_0^{\text{edit}} = Z_0^{\text{src}} = X^{\text{src}}$, $Z_1^{\text{edit}} = Z_1^{\text{tar}} = X^{\text{tar}}$. Consequently, the evolution of the entire editing flow path, *i.e.*, $Z_t^{edit}$ from $t = 1$ to $t = 0$ achieve the transition from source video to the targeted editing video. Take the derivative with respect to $t$ of both sides give us the editing flow ODE,

$$dZ_t^{\text{edit}} = V(Z_t^{\text{edit}}, t)dt, \tag{4}$$

$$V(Z_t^{\text{edit}}, t) = \frac{\partial Z_t^{\text{edit}}}{\partial t} = V(Z_t^{\text{tar}}, t) - V(Z_t^{\text{src}}, t), \tag{5}$$

where $V_t^{\text{edit}}$ is the direct editing velocity flow that drives the desired editing at each step. As indicated in Eq. (5) and illustrated in Fig. 1, the constructed editing path can be understood as the composition of the source flow path of $Z_t^{\text{src}}$ and target flow path $Z_t^{\text{tar}}$, where former drive the $Z_t^{\text{edit}}$ away from source video and later one pull it toward the target video, leading to the composited estimation of the editing direction. Note that we estimate the velocity $V_t$ with pre-trained T2V flow model $v_\theta$, specifically, we permute the source video as the rectified flow schedule, $z_t^{\text{src}} = (1 - t)x^{\text{src}} + t\epsilon_t$, where $\epsilon_t \sim N(0, I)$ and the source flow is then estimated with $V_t^{\text{src}} = v_\theta(z_t^{\text{src}}, t, c_{\text{src}})$. Given that $X^{\text{tar}}$ is unknown until we accomplish the editing, we rearrange Eq. (3) to obtain:

$$Z_t^{\text{tar}} = Z_t^{\text{edit}} + Z_t^{\text{src}} - Z_0^{\text{src}}. \tag{6}$$

Since we start from $Z_1^{\text{edit}} = Z_0^{\text{src}}$, the $V_t^{\text{tar}}$ can also be estimated via Eq. (6) along the editing path evolving from $t = 1$ to $t = 0$. Note that from Eq. (6) can be observed that $Z_t^{\text{tar}}$ and $Z_t^{\text{src}}$ contain roughly the same noise constituent, therefore the editing flow $V_t^{\text{edit}}$ obtained from velocity estimation $V_t^{\text{tar}}$ are also the similar case and thus the obtained $V_t^{\text{src}}$ via Eq. (5) remove roughly the same noise component, finally encompasses the difference only between the clean video predictions. In other words, *the estimated editing velocity $V_t^{edit}$ directly describes the desired change of the source video at each step in the data space* as shown in Fig. 5.

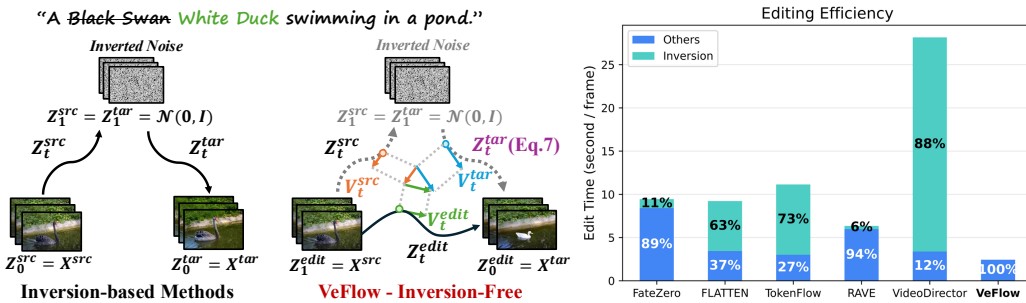

Figure 1: *Left*: **Video Editing Flow Illustration**. VEFLOW directly estimates the editing flow at each step to transform the source video into the target video within data space, eliminating the need for inversion. *Right*: **Editing Efficiency Comparison**. By avoiding explicit inversion, our approach achieves significantly improved editing efficiency.

**Attention-guided Flow Masking**. The direct video editing flow in Eq. (5) estimates the desired alteration upon the source video at each step based on the source prompt and editing prompt. This enables us to precisely control the editing region over the source video. Specifically, we leverage the masks derived from cross-attention maps of $v_\theta(z_t^{\text{src/tar}}, t, c_{\text{src/tar}})$ to suppress undesired changes. Suppose the video latent in the shape of $P^{f \times h \times w}$, where $f, h, w$ represent the frame, height, and width, and the textual prompt tokens in the length of $s$, the flow model $v_\theta$ injects the prompt condition $c^{\text{tar/tar}}$via cross-attention as[1]:

$$M = \text{Softmax}\left(\frac{QK^\top}{\sqrt{d}}\right), \quad \text{where} \quad \text{Attn}(Q, K, V) = M \cdot V, \tag{7}$$

$d$ is the latent projection dimension of the keys and queries, and $M \in \mathbb{R}^{fhw \times s}$ represents the text-video cross-modal attention map. The cell $M_{ij}$ defines the weight of the value of the $j$-th token on the pixel $i$. We exploit the cross-attention map of the key editing tokens to localize the desired editing region and keep the remaining region as untouched as possible. Specifically, taking $v_\theta(z_t^{\text{src}}, t, c_{\text{src}})$ as an example, we obtain the binary editing region $\mathcal{M}^{\text{src}}$ by thresholding the attention map $M^{\text{src}}$ with threshold of $\gamma$,

$$\mathcal{M}^{\text{src}} = \mathbb{I}\left(\frac{\hat{M}^{\text{src}} - \min(\hat{M}^{\text{src}})}{\max(\hat{M}^{\text{src}}) - \min(\hat{M}^{\text{src}})} > \gamma\right), \quad \text{where} \quad \hat{M}^{\text{src}} = \frac{1}{|K_s|}\sum_{j \in K_s} M_{i,j}^{\text{src}}, \tag{8}$$

and $K_s$ denotes the token index span of the source editing keywords. However, we found that the binary mask produced in this way tends to be scattered and fails to fully capture the editing regions, as shown in Fig. 5. To this end, we utilize a series of morphological operations to infer the complete editing mask. Specifically, we first take the close operation to connect the nearby elements, and then, an open operation is applied to suppress the remaining isolated elements. Finally, a dilation operation is applied to expand the region to a certain degree:

$$\mathcal{M}'^{\text{src}} = \text{Dilation}\Big(\text{Open}\big(\text{Close}(\mathcal{M}^{\text{src}}, K_c), K_o\big), K_d\Big), \tag{9}$$

where $K_c, K_o, K_d$ are the kernel sizes of the close, open, and dilation operations, respectively. Considering that there may be significant shape deformation after editing, we take the union of the source and target editing region masks to obtain the final editing mask $\mathcal{M}^{\text{edit}} = \mathcal{M}'^{\text{src}} \cup \mathcal{M}'^{\text{tar}}$. The $\mathcal{M}^{\text{edit}}$ represents the region where editing occurs, including the source subject disappearing and the target subject emerging, thereby is utilized to mask $V_t^{\text{edit}}$ within the certain editing step range of $[t_1, t_2]$ to suppress the undesired background change.

**Decoupled Flow Modulation**. The editing path construction shown in Eq. (5) can be understood as a composite effect of the source disappearance (driven by $-V_t^{\text{src}}$) and target emergence (driven by $V_t^{\text{tar}}$) as illustrated in Fig. 5. However, we observe that these two altering directions may conflict sometimes, leading to insufficient editing results (*e.g.*, partially emerged target subject). Empirically, we found that it occurs when the editing source and target object share similar semantics, where the source flow attempts to erase it from the source video, while the target flow enforces it to emerge, leading to a conflict editing force and impairing the editability.

---

[1]For clarity, $t$ is omitted here.

To this end, we propose to decouple these two flow directions to mitigate the potential conflict. Specifically, we modify the original editing flow as:

$$V_t'^{\text{edit}} = V_t^{\text{tar}} - \text{Proj}\left(V_t^{\text{tar}}, V_t^{\text{src}}\right), \tag{10}$$

where $\text{Proj}(\cdot, \cdot)$ represent the projection operation, and is defined as:

$$\text{Proj}\left(V_t^{\text{tar}}, V_t^{\text{src}}\right) = \left(\frac{V_t^{\text{tar}} \cdot V_t^{\text{src}}}{\|V_t^{\text{src}}\|}\right) V_t^{\text{src}}. \tag{11}$$

Here, $\cdot$ denotes the dot product, and $\|\cdot\|$ is the vector norm. This projection allows the target emergence flow to counteract the erasing force from the source disappearance flow, focusing exclusively on the unique transformation required for the source-to-target conversion. After removing this conflicting component, we amplify the resulting flow $V_t'^{\text{edit}}$ by a coefficient $w > 1$ to strengthen the target editing force. Combining this with our masking strategy yields the final editing flow applied at each timestep:

$$V_t^{\text{edit}} = \begin{cases} w \cdot \mathcal{M}_t^{\text{edit}} \otimes V_t'^{\text{edit}}, & t \in [t_1, t_2], \\ V_t^{\text{edit}}, & \text{otherwise.} \end{cases} \tag{12}$$

The complete procedure is summarized in Alg. 1.

---

**Algorithm 1** Video Editing with VEFLOW

---

1: **Input:** Source video $X^{\text{src}}$, text prompts $(c^{\text{src}}, c^{\text{tar}})$, timesteps $\{t_i\}_{i=0}^T$, total steps $n_{\max}$, editing interval $[t_1, t_2]$, flow model $v_\theta$, modulation coefficient $w$
2: **Output:** Edited video $X^{\text{tar}}$ corresponding to $c^{\text{tar}}$
3: **Initialize:** $Z_{t_{n_{\max}}}^{\text{edit}} = X^{\text{src}}$
4: **for** $i = n_{\max}$ **down to** $1$ **do**
5:     $N_{t_i} \sim \mathcal{N}(\mathbf{0}, \mathbf{I})$
6:     $Z_{t_i}^{\text{src}} \leftarrow (1 - t_i)X^{\text{src}} + t_i N_{t_i}$
7:     $Z_{t_i}^{\text{tar}} \leftarrow Z_{t_i}^{\text{edit}} + Z_{t_i}^{\text{src}} - X^{\text{src}}$
8:     **if** $t_i \in [t_1, t_2]$ **then**
9:        $\mathcal{M}_t^{\text{edit}} \leftarrow \text{AFM}(v_\theta, c^{\text{src}}, c^{\text{tar}})$
10:       $V_{t_i}'^{\text{edit}} \leftarrow \text{DFM}(Z_{t_i}^{\text{tar}}, Z_{t_i}^{\text{src}})$
11:       $Z_{t_{i-1}}^{\text{edit}} \leftarrow Z_{t_i}^{\text{edit}} + w \cdot (t_{i-1} - t_i) \cdot \mathcal{M}_t^{\text{edit}} \otimes V_{t_i}'^{\text{edit}}$
12:     **else**
13:        $V_{t_i}^{\text{edit}} \leftarrow V^{\text{tar}}(Z_{t_i}^{\text{tar}}, t_i) - V^{\text{src}}(Z_{t_i}^{\text{src}}, t_i)$
14:       $Z_{t_{i-1}}^{\text{edit}} \leftarrow Z_{t_i}^{\text{edit}} + (t_{i-1} - t_i) \cdot V_{t_i}^{\text{edit}}$
15:     **end if**
16: **end for**
17: **Return:** $X^{\text{tar}} = Z_0^{\text{edit}}$

---

## 3 EXPERIMENTS

### 3.1 EXPERIMENTAL SETTINGS

**Datasets and Baselines**. We evaluate our inversion-free text-to-video editing framework with the real-world videos sourced from existing benchmarks (*e.g.*, DAVIS) and downloaded from the internet. We compared our approach with several state-of-the-art text-driven video editing methods: FateZero Qi et al. (2023), FLATTEN Cong et al. (2023), TokenFlow Geyer et al. (2023), RAVE Kara et al. (2024), and VideoDirector Wang et al. (2025). For all these baseline methods, we follow the default settings provided in their official GitHub repositories. More detailed experimental settings for our method are provided in Sec. A.3.

**Implementation Details**. We adopt the open-sourced flow-based video generation model - Wan 2.1 Wan et al. (2025) as the base model to implement VEFLOW. We perform inversion-free video editing with 50 denoising steps, and skip the first 5 editing steps (*i.e.*, the maximum editing steps $n_{\max} = 45$) to balance structural consistency with editing freedom. We implement attention-guided flow masking (AFM) by extracting the cross-attention masks from the attention maps of all DiT blocks. The cross-attention threshold was set to $\gamma = 0.2$. The kernel size used in the mask post-process was set $K_c = 3, K_o = 5, K_d = 5$. The applied editing step range is set to $[t_1, t_2] = [5, 25]$. All experiments for VEFLOW were run on a single NVIDIA L40 GPU unless otherwise specified.

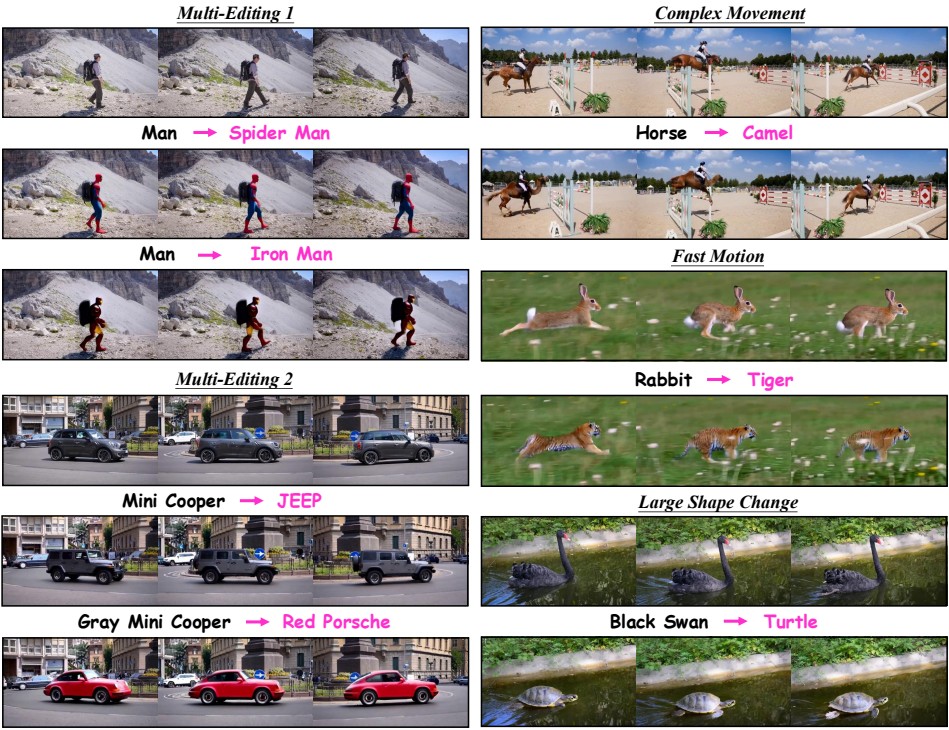

Figure 2: **Visual Example of VEFLOW**. Our approach exhibits superior performance in video editing under various conditions. These include replacing a subject with multiple different ones, editing videos of complex movement and fast motion, and handling cases involving significant shape changes. Best viewed zoomed in.

Table 1: **Quantitative Comparison**. Our method demonstrates superior performance in background preservation, editing alignment, motion fidelity, and frame quality.

| Methods | Background Preservation | | | | T2V Alignment | Frame Quality | Motion Fidelity | |
|---|---|---|---|---|---|---|---|---|
| | PSNR↑ | LPIPS↓ | MSE↓ | SSIM↑ | CLIP-T ↑ | PickScore ↑ | WarpSSIM↑ | Temporal Flicker↑ |
| FateZero | 15.82 | 232.72 | 316.63 | 57.45 | 31.26 | 20.72 | 60.51 | 91.92 |
| FLATTEN | 14.76 | 301.03 | 382.83 | 55.84 | 30.32 | 19.85 | 68.61 | 94.64 |
| TokenFlow | 21.62 | 130.12 | 81.86 | 75.62 | 29.90 | 20.83 | 67.58 | 96.38 |
| RAVE | 14.92 | 328.74 | 562.91 | 58.35 | 30.42 | 20.95 | 65.68 | 95.69 |
| VideoDirector | 16.58 | 240.38 | 258.29 | 57.82 | 30.19 | 20.50 | 63.40 | 94.62 |
| Ours | **22.44** | **78.95** | **64.32** | **85.15** | **31.94** | **21.30** | **70.32** | **96.71** |

## 3.2 MAIN RESULTS

**Quantitative Evaluation**. We conduct a comprehensive quantitative comparison between our method, VEFLOW , and other state-of-the-art text-driven video editing approaches. Our evaluation focuses on four key aspects: background preservation, text-video alignment, frame quality, and motion fidelity. To assess background preservation, we first use SAM2 Ravi et al. (2024) to segment the edited subject. We then compute PSNR, LPIPS Zhang et al. (2018), MSE, and SSIM Wang et al. (2004) metrics exclusively within the unedited regions between the source and edited videos. For text-video alignment and frame quality, we report the average CLIP Radford et al. (2021) score and PickScore Kirstain et al. (2023) across all frames. While for the motion fidelity, we utilize WarpSSIM to evaluate the motion consistency, which calculates the average SSIM between the final edited video and the source video warped using RAFT Teed & Deng (2020) optical flow. We further utilize the temporal flicker from VBench Huang et al. (2024) to detect the potential motion defects after editing. As shown in Tab. 1, our proposed method significantly outperforms existing state-of-the-art approaches. It achieves superior alignment with user edit intentions, higher output frame quality, and more natural motion preservation. Notably, VEFLOW demonstrates a substantial improvement in background preservation while maintaining strong editing performance, thereby enhancing its practicality for real-world applications.

**Qualitative Evaluation**. We perform video editing with VEFLOW for various real-world videos, and showcase the editing results in Fig. 2. In summary, our method exhibits superior editing capability in various challenging situations, including consistent subject replacement, editing videos with complex scenes and fast motion, and handling cases involving significant shape deformation. For example, we can easily change the person into a different identity, such as "Spider-Man" and "Iron Man". It also performs well when the video contains complicated scenes and complex or fast movements. Moreover, unlike some methods Qi et al. (2023) that need to take a different way to process the editing of attribute change and large shape alteration, our approach can deal with various situations, even with a significant shape deformation (see example of swan). We also provide the visual comparison with the existing methods in Fig. 3. It can be observed that our approach not only precisely achieves the desired change in the source video and yields more satisfactory editing results, but also maintains high background consistency before and after the editing.

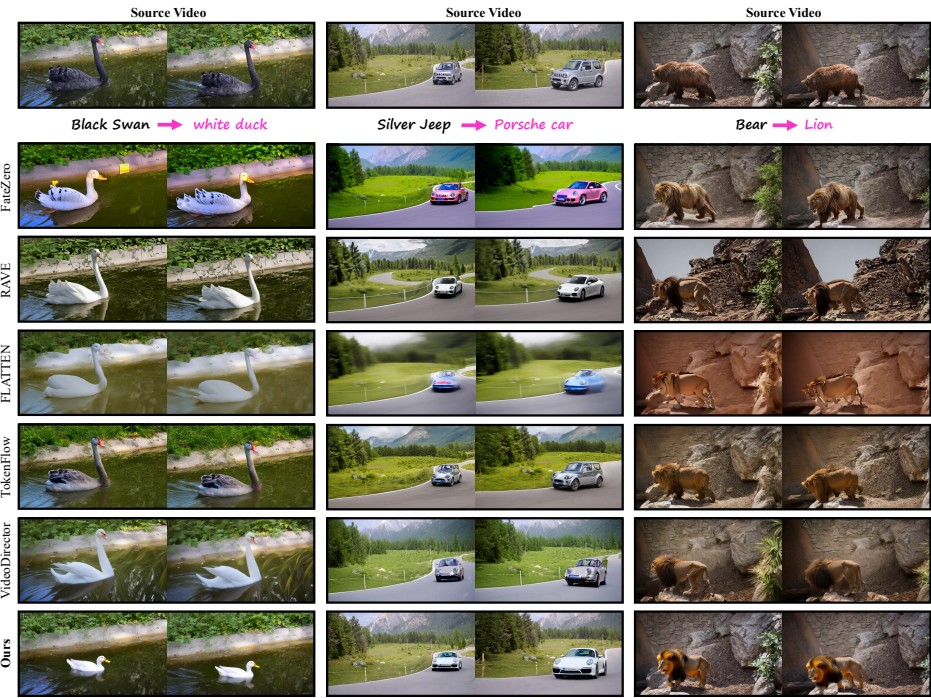

Figure 3: **Qualitative Comparison**. Our method outperforms previous methods across diverse editing cases, demonstrating superior editing alignment and background preservation. Best viewed zoomed in.

**Editing Efficiency**. We compare the per-frame editing time of different methods in Fig. 1 with a single NVIDIA H800. Our approach demonstrates superior editing efficiency, consuming significantly less time than existing methods. This efficiency gain stems from our novel direct editing flow design, which eliminates the inversion step required by prior work. By contrast, the inversion procedure may incur considerable time cost within the conventional "invert-then-edit" approaches. For instance, methods like VideoDirector Wang et al. (2025) rely on techniques such as Null-Text Optimization Mokady et al. (2023) to reduce reconstruction errors during inversion, involving computationally expensive gradient updates that substantially increase processing time, taking almost 88% of the time of the whole editing pipeline. In contrast, our method performs edits directly at each timestep without any optimization, achieving markedly higher efficiency.

### 3.2.1 ABLATION STUDY

**Component Effectiveness**. We evaluate the effectiveness of our key designs in Fig. 4 and Tab. 2. It is evident that the proposed attention-guided flow masking (AFM) module effectively suppresses unintended background changes (*e.g.*, the bench in Fig. 4). However, due to conflicts between the subject disappearing force incurred by the source flow and the emerging force of the target flow, the baseline with only AFM yields suboptimal editing results, leading to unclear object appearance (*e.g.*, the unclear robot contour). By incorporating the decoupled flow modulation (DFM) module

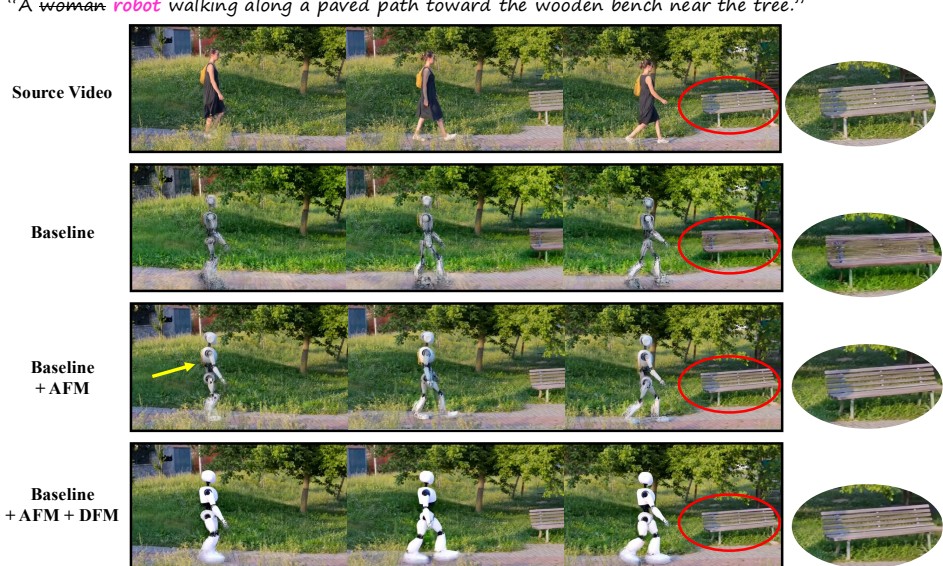

Figure 4: **Ablation on Key Components**. Compared to the baseline, the proposed attention-guided flow masking (AFM) module effectively preserves the background, while the decoupled flow modulation (DFM) module enhances the overall editing performance.

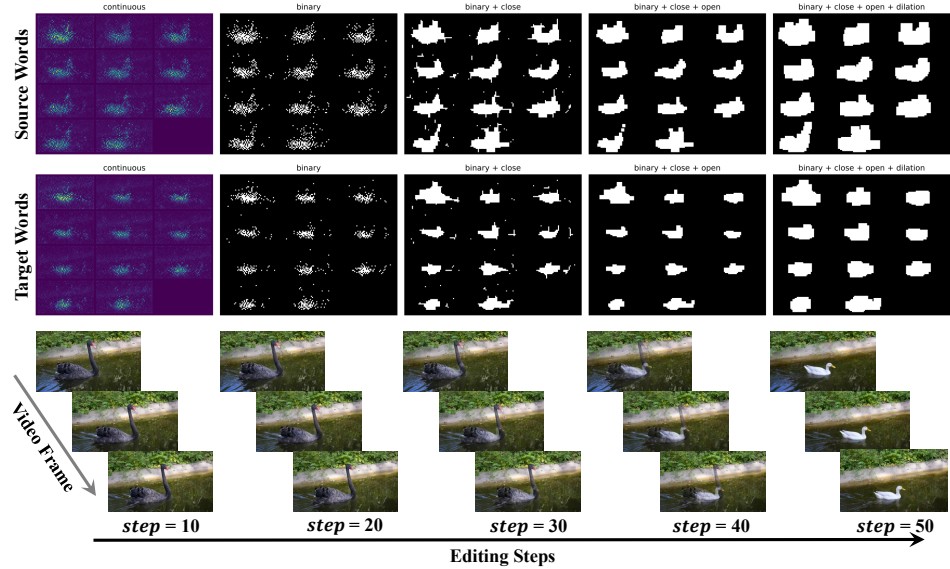

Figure 5: **Analysis on the Flow Masking**. *Top*: The procedure of the source and target editing region mask generation via AFM at editing steps of 10. *Bottom*: The transition from the source video to the target video within the data space driven by VEFLOW .

to mitigate such conflicts via flow projection and modulation, the target object is edited more effectively and exhibits a clearer identity. This is further supported by a quantitative improvement in text-video alignment, as shown in Tab. 2.

**Analysis on Flow Masking**. We employ the cross-attention mask to localize the editing region, which is used to mask the unintended editing flow in the background region. As shown in Fig. 5, the initial attention maps for the source ("black swan") and target keywords "white duck" are often scattered and diffuse, especially in early denoising steps, failing to provide precise localization. Applying our proposed morphological post-processing yields more compact and clearly defined masks, which align accurately with the intended source and target editing regions.

Table 2: **Component Effectiveness Ablation**. The proposed attention-guided flow masking module contributes significantly to the background preservation over the baseline, while he flow modulation further improves the alignment between the edited video and the editing prompt.

| Methods | Background Preservation | | | | T2V Alignment |
|---|---|---|---|---|---|
| | PSNR↑ | LPIPS↓ | MSE↓ | SSIM↑ | CLIP-T ↑ |
| Baseline | 20.47 | 97.04 | 83.89 | 81.08 | 31.31 |
| Baseline + *AFM* | 21.56 | 80.99 | **58.42** | 84.43 | 31.53 |
| Baseline + *AFM* + *DFM* | **22.44** | **78.95** | 64.32 | **85.15** | **31.94** |

**Analysis on Flow Modulation**. As the intermediate edited results shown in Fig. 5, the source disappearing and target emerging editing directives coexist within the video editing flow. This coexistence can lead to conflicts, particularly when the source and target objects share high-level semantics but have distinct appearances, resulting in insufficient editing results. To address this issue, we propose the DFM module to mitigate such conflict via the flow projection and modulation. Fig. 6 demonstrates the effect of these operations. When the editing flow is applied directly without projection (*i.e.*, w/o Proj.), the result exhibits significant artifacts, such as the incomplete hand and leg of Iron Man. This occurs because the source flow $V_t^{\text{src}}$ tries to erase these parts (original man's) while the target flow $V_t^{\text{tar}}$ simultaneously attempts to generate them (Iron Man's), creating a direct conflict. Projecting the target flow (*i.e.*, $w = 1$) to obtain an orthogonal editing direction that eliminates the conflicting component, yielding a clearer rough contour. By further modulating the projected flow with the coefficient $w > 1$ to strengthen the target editing direction, it ultimately leads to the desired satisfactory editing results. Empirically, $w \in [1.5, 2.0]$ works well, while excessively larger values can result in over-editing.

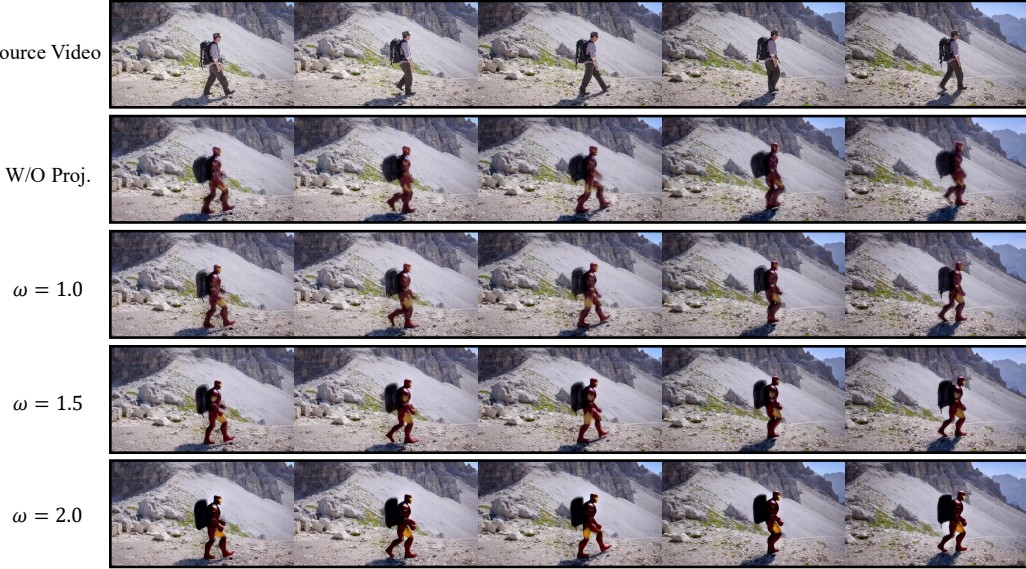

Figure 6: **Analysis on the Decoupled Flow Modulation**. Starting from the baseline equipped with the AFM module, the flow projection and modulation are incrementally incorporated. Best viewed zoomed in.

## 4 CONCLUSION

We introduce VEFLOW, an inversion-free, training-free text-driven video editing framework based on the flow-based generative model. By deriving the video editing flow that directly transforms the source video to the target edited results, it enables highly efficient video editing within the data space without inversion. With the further developed attention-guided flow masking and decoupled flow modulation module, it achieves superior background preservation and editability across diverse scenarios. Extensive experiments on real-world videos demonstrate that VEFLOW significantly outperforms existing state-of-the-art methods.

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

# A APPENDIX

## A.1 RELATED WORKS

**Text-to-Video Generation**. Text-to-video (T2V) generation have went through remarkable advancement over the past few years, largely driven by the development of diffusion models Ho et al. (2020); Song et al. (2021). Building on advanced text-to-image (T2I) generative models Rombach et al. (2021); Chen et al. (2023b); Podell et al. (2023), early works Guo et al. (2024); Zhou et al. (2022); Wang et al. (2024; 2023) commonly inflate the pre-trained image diffusion model Rombach et al. (2021) into the video diffusion model by incorporating extra temporal layers (*e.g.*, temporal convolution or temporal attention). For instance, Animatediff Guo et al. (2024) extended a pre-trained T2I model by introducing an additional motion module to learn temporal dynamics through fine-tuning on video data, achieving preliminary text-to-video generation. Subsequent efforts, such as large-scale video pre-trained diffusion models including VideoCrafter Chen et al. (2023a; 2024) and CogVideoX Yang et al. (2024), demonstrated impressive video quality by leveraging extensive video datasets for pre-training. More recently, with the emergence of flow-based generative models Esser et al. (2024); Liu et al. (2022), researchers have begun to explore flow-based approaches for video generation. Unlike diffusion-based methods, models such as Chen et al. (2025) are trained using the more stable flow-matching objective, resulting in superior video generation quality. Our approach builds upon the latest flow-based video generation model, Wan model Wan et al. (2025), to achieve high-quality video editing.

**Text-Driven Video Editing**. Early efforts in video editing primarily extended text-to-image (T2I) models Ho et al. (2020); Rombach et al. (2021); Song et al. (2021); Podell et al. (2023) directly to videos. To address the domain gap between images and videos, these methods focused heavily on improving temporal consistency. For instance, TokenFlow Geyer et al. (2023) decouples the task into appearance alteration and motion transfer: it first performs DDIM inversion Song et al. (2020) to extract the source video's noise latents, then propagates temporal correspondences to ensure the edited video maintains the original motion while reflecting appearance changes. Similarly, FLAT-TEN Cong et al. (2023) edits content via a T2I model and employs optical-flow guided attention to preserve source motion patterns. RAVE Kara et al. (2024) utilizes ControlNet Zhang et al. (2023) with random latent shuffling to enhance temporal coherence. Recently, methods have shifted toward leveraging native text-to-video (T2V) models Yang et al. (2024) for better inherent temporal consistency. For example, VideoDirector Wang et al. (2025) edits directly on T2V models and introduces spatial-temporal decoupled guidance to further improve coherence. However, despite achieving impressive results, these methods still follow the "invert-then-edit" paradigm, which suffers from two key limitations: considerable time cost incurred by the iterative inversion process, and noticeable unintended alteration caused by the inherent reconstruction errors. In contrast, our method bypasses inversion entirely by reformulating video editing as a direct editing ordinary differential equation (ODE) flow, enabling both precise and efficient editing.

## A.2 ADDITIONAL PRELIMINARY

**Rectified Flow**. Rectified Flows (RFs) Liu et al. (2022); Esser et al. (2024) are a particular type of flow matching models Lipman et al. (2022). It constructs a special ODE path such that the marginal at time $t$ corresponds to a linear interpolation between $X_0$ and $X_1$. Specifically, let $x_0 \sim X_0$ denote a data sample drawn from the true data distribution, and $x_1 \sim X_1$ denote a noise sample from a Gaussian distribution, the rectified flow framework, which defines the "noised" data $z_t$ as $z_t = (1-t)x_0 + tx_1$ for $t \in [0, 1]$. Then a neural network model parameterized with $\theta$ is trained to directly regress the velocity field $v_\theta(z_t, t, c)$ by minimizing the Flow Matching objective:

$$\mathcal{L}(\theta) = \mathbb{E}_{t, x_0 \sim X_0, x_1 \sim X_1} \|\hat{v} - v_\theta(x_t, t, c)\|^2, \tag{13}$$

where the target velocity field is $\hat{v} = x_1 - x_0$, $c$ is the condition. Modern flow-based image or video generative models are trained with vast text-image/video pairs, $(c, x_0) \in (C, X_0)$, to allow for generating the samples that have the aligned content specified by the textual prompt. One desired property of the recitified flow models is that their sampling paths are relatively "straight" Liu et al. (2022), which allows using a small number of discretization steps when solving the ODE with the numerical ODE solver Karras et al. (2022).

**Diffusion Model**. Diffusion models (DMs) models the data distribution by formulating the transformation from a Gaussian distribution to the data distribution as a Markov diffusion process. Basically,

we consider the latent diffusion models (LDMs), which define a forward process to diffuse a clean image latent with a Gaussian noisy latent as:

$$\boldsymbol{z}_t = \sqrt{\alpha_t}\boldsymbol{z}_0 + \sqrt{1 - \alpha_t}\epsilon_t, \text{ where } \epsilon_t \sim \mathcal{N}(\boldsymbol{0}, \boldsymbol{I}), \tag{14}$$

where $z_0$ is the clean image latent encoded by the VAE encoder $\mathcal{E}(\cdot)$. During training, given the noisy latent $z_t$ and condition $c$ such as text, the diffusion model $\epsilon_\theta$ is encouraged to predict the noise $\epsilon_t$ at step t with training objective:

$$\mathcal{L}(\theta) = \mathbb{E}_{\mathcal{E}(x),\epsilon\sim\mathcal{N}(\boldsymbol{0},\boldsymbol{I}),t\sim\mathcal{U}(1,T)} \left[ \|\epsilon_t - \epsilon_\theta\left(\boldsymbol{z}_t, c, t\right)\|_2^2 \right]. \tag{15}$$

After training, the sample is generated by the reverse process to denoise from a Gaussian noise sample with the noise predicted via $\epsilon_\theta$. Specifically, we initialize the latent $z_T = \mathcal{N}(0, I)$ and iterative update $z_{t-1}$ from $z_t$ with the predicted noise $\epsilon_\theta$. To facilate more accurate estimation of $\epsilon_\theta$, classifier-free guidance (CFG) Ho & Salimans (2022) is usually applied as:

$$\hat{\epsilon}_\theta = \epsilon_\theta(z_t, c, t) + \alpha[\epsilon_\theta(z_t, c, t) - \epsilon_\theta(z_t, \phi, t)], \tag{16}$$

where $\alpha$ is the guidance scale, $\phi$ represents null-text or a negative prompt.

**DDIM Sampling and Inversion**. Traditional diffusion model generation involves thousands of denoising operations along the Markov diffusion chain. In contrast, DDIM introduces an efficient sampling strategy that reduces this to merely tens of steps. The denoising transition from $z_t$ to $z_{t-1}$ in DDIM is formulated using the predicted noise $\epsilon_\theta(z_t)$ as:

$$\boldsymbol{z}_{t-1} = \sqrt{\alpha_{t-1}} \left( \frac{\boldsymbol{z}_t - \sqrt{1 - \alpha_t}\epsilon_\theta(\boldsymbol{z}_t)}{\sqrt{\alpha_t}} \right) + \sqrt{1 - \alpha_{t-1}}\epsilon_\theta(\boldsymbol{z}_t). \tag{17}$$

The DDIM formulation also enables a deterministic inversion process that maps a cleaner image latent back to a noisier state. Specifically, we derive the transformation expressing $z_t$ in terms of $z_{t-1}$, and then shift the time indices from $(t, t-1)$ to $(t+1, t)$. This yields the DDIM inversion formulation:

$$\boldsymbol{z}_{t+1} = \sqrt{\alpha_{t+1}} \left( \frac{\boldsymbol{z}_t - \sqrt{1 - \alpha_t}\epsilon_\theta(\boldsymbol{z}_t)}{\sqrt{\alpha_t}} \right) + \sqrt{1 - \alpha_{t+1}}\epsilon_\theta(\boldsymbol{z}_t). \tag{18}$$

This inversion process transforms the clean image latent $z_0$ into the noisy latent $z_t$ through iterative updates, ultimately reaching the Gaussian noise latent $z_T$. However, since $\epsilon_\theta(z_{t+1})$ cannot be computed without knowing $z_{t+1}$, common practice approximates $\epsilon_\theta(z_{t+1})$ with $\epsilon_\theta(z_t)$, which is valid only when the time interval is sufficiently small. This approximation not only limits the ability to fully recover original content during denoising but also requires considerable inversion steps.

DDIM inversion offers a practical framework for text-driven content editing by first inverting source data into a feature-preserving Gaussian noise latent, which is then denoised with an editing prompt to achieve desired modifications. This approach, established in image editing, has been extended to video editing in early works Liu et al. (2024); Qi et al. (2023); Cong et al. (2023).

**Null-Text Optimization**. Due to the approximation introduced during DDIM inversion, it leads the deviations between the inverted noise latent and the ideal no-bias noise latent that reconstruct the exact source input. To address this issue, Mokady et al. (2023) introduced a null-text embedding optimization technique to reduce the inversion error. Specifically, it optimize a step-wise null-text embedding $\phi_t$ after DDIM inversion, which is achieve via the optimization of $\phi_t$ at each step with the objective as:

$$\mathcal{L}(\phi_t) = \|z_{t-1}^* - z_{t-1}\|_2^2, \tag{19}$$

where $z_t$ and $z_t^*$ represent the latents from denoising and DDIM inversion, respectively. During the re-denoising with the inverted noise latent, such optimized null-text embedding $\phi_t$ is used via Eq. (16) to serve as a compensation for DDIM inversion biases, enhancing both reconstruction and editing quality. Note that VideoDirector Wang et al. (2025) extend such optimization techniques into the video domain, although improve the reconstruction error, it incurs considerable time cost due to the iterative optimization of each step, which hurt the editing efficiency significantly.

A.3    EXTENDED IMPLEMENT DETAILS

**Detailed Experiment Setup**. We use Wan2.1 Wan et al. (2025) from the official Huggingface repository as our base model. Specifically, we employ the `Wan2.1-T2V-1.3B-Diffusers` variant. Source videos are sampled and rescaled to a resolution of $832 \times 480$ with 41 frames. We apply classifier-free guidance Ho & Salimans (2022) with scales of 1 and 7.5 for estimating $V(Z_t^{\text{src}}, t)$ and $V(Z_t^{\text{tar}}, t)$, respectively. For flow estimation during sampling, we use the discrete Euler scheduler with a timestep shift scale (`sigma_shift`) set to 5 as suggested by the Wan2.1 official repo.

**Editing Efficiency Evaluation Details**. To ensure a fair comparison in editing efficiency evaluation, we adaptively modified all methods to support the same resolution as our approach. Beyond this resolution adjustment, no additional optimization techniques were applied, and all experiments were conducted using the default settings specified in the respective official repositories. During preliminary testing, we observed that VideoDirector consistently encountered out-of-memory issues on L40 GPUs with 48GB of memory due to gradient computations in its inversion process. Consequently, we evaluated all methods using a single H800 GPU with 80GB of memory to maintain consistent experimental conditions.

A.4    ADDITIONAL EXPERIMENTAL RESULTS

**When to Apply AFM and DFM**. One crucial design choice of VEFLOW is to choose the proper editing step range to impose the AFM and DFM module. Since the denoising flow is known to determine the overall structure and layout in the early stage Hertz et al. (2022), we applied our approach at the early stage to avoid significant global alteration and determine which specific step to stop applying. The ablated experiment results are summarized in Fig. 7. We found it is not practical to apply AFM and DFM throughout all the editing steps. Despite achieving the desired editing, applying the AFM and DFM at all editing steps leads to the edited goat looking unnatural with the surrounding content. To demonstrate this, we visualize the attention map of $\hat{M}^{\text{src}}$ and $\hat{M}^{\text{tar}}$ at different editing step stage in Fig. 7. It can be observed that in the late stage of the editing flow, the attention region of both source and target keywords spreads out across the frames, making it unsuitable to extract the flow masking for AFM. In fact, the previous studies Hertz et al. (2022); Kulikov et al. (2024) reveal that the late stage of the denoising primarily focuses on refining some subtle high-frequency details, like making the edited subject blend naturally with the original video unedited region. Therefore, we choose not to apply our approach at the late stage of editing. Empirically, we found the $t_2 \in [25, 30]$ strike a good balance between the editability and the fidelity. In our main paper, we choose $[t_1, t_2] \in [5, 25]$.

**Effect of Flow Projection**. One might question whether the effectiveness of DFM stems primarily from the amplification via $w > 1$, rather than from the projection mechanism that mitigates conflicting edits. To address this concern, we conduct an ablation study (results summarized in Fig. 9). The results show that while directly amplifying the editing flow improves performance to some extent, using the same coefficient $w$ on the *projected* flow yields more effective editing alterations (particularly notable in the first frame). This demonstrates that the conflicting editing force from the source flow indeed impairs the target editing flow's effectiveness, and that projection successfully mitigates this issue.

**More Qualitative Comparison**. We provide more visual editing results comparison with other methods in Fig. 10. We found that some methods exhibit very unstable editing results, particularly VideoDirector Wang et al. (2025). For example, in the case of editing from cow to sheep, VideoDirector produces desired editing results that align with the editing prompt, while preserving the background decently. However, in the case of black swan and jogger, it leads to significant background alternation as well as inferior editing results. By contrast, our approach delivers consistent superior editing performance across various scenarios, including aligned editing and well background preservation.

**More Visual Edit Results**. We provide more video editing results of our method in Fig. 11. It can be observed that our method exhibits superior editing performance in various situations, even for subjects on a small scale (*e.g.*, seagull), it still precisely locates the object we want to modify and correctly alters the content. It is worth noting that we observe some interesting editing results. As shown in the case of editing the rhinoceros in Fig. 11, our approach successfully modifies the

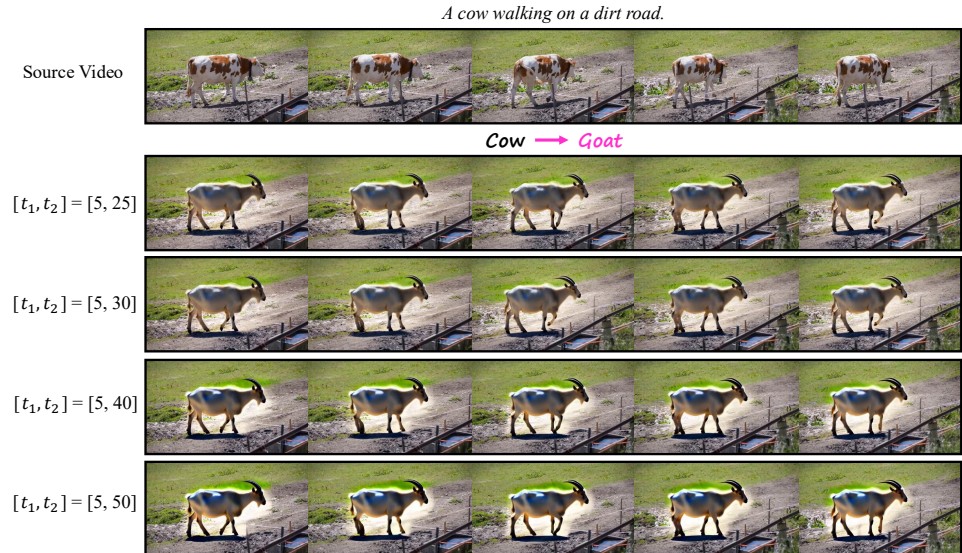

Figure 7: **Ablation on the Applied Editing Steps**.It requires leaving the late stage of the editing flow unattended to harmonize the edited results.

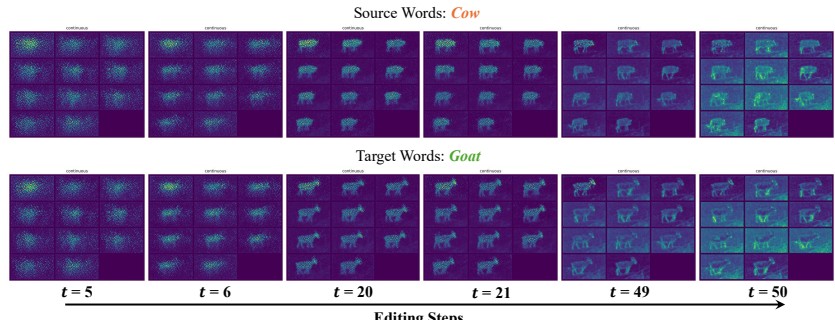

Figure 8: **Cross-Attention Map of Different Editing Steps**. The source and target cross-attention map at different editing steps.

rhinoceros into an elephant according to the user's intention. However, the edited elephant seems to overlook the tree and does not walk behind the tree as in the source video. In other words, it falls short in recognizing the occlusion, but it indeed achieves the user's editing request. We think it finally depends on the user's specific requirement to determine whether this case is a failure case or not.

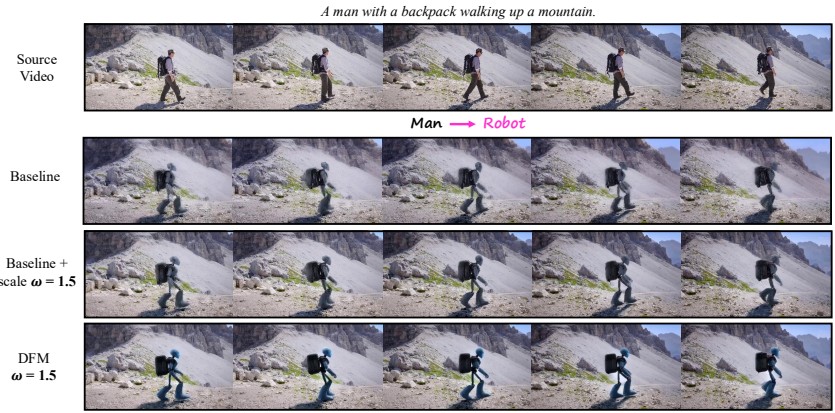

Figure 9: **Ablation of Flow Projection**. The projection mitigates the conflict component within the editing flow, providing a more concentrated editing direction.

### A.5  USE OF LARGE LANGUAGE MODELS (LLMS)

The author(s) used a Large Language Model, including GPT-4o-mini and DeepSeek AI, solely to assist in polishing and improving the fluency of pre-written paragraphs. The model was used for tasks such as rephrasing for clarity, correcting grammar, and ensuring stylistic consistency. All conceptual development, research ideation, analysis, and original writing were performed by the author(s). The LLM was used as an editorial tool and did not contribute to the intellectual substance of the work.

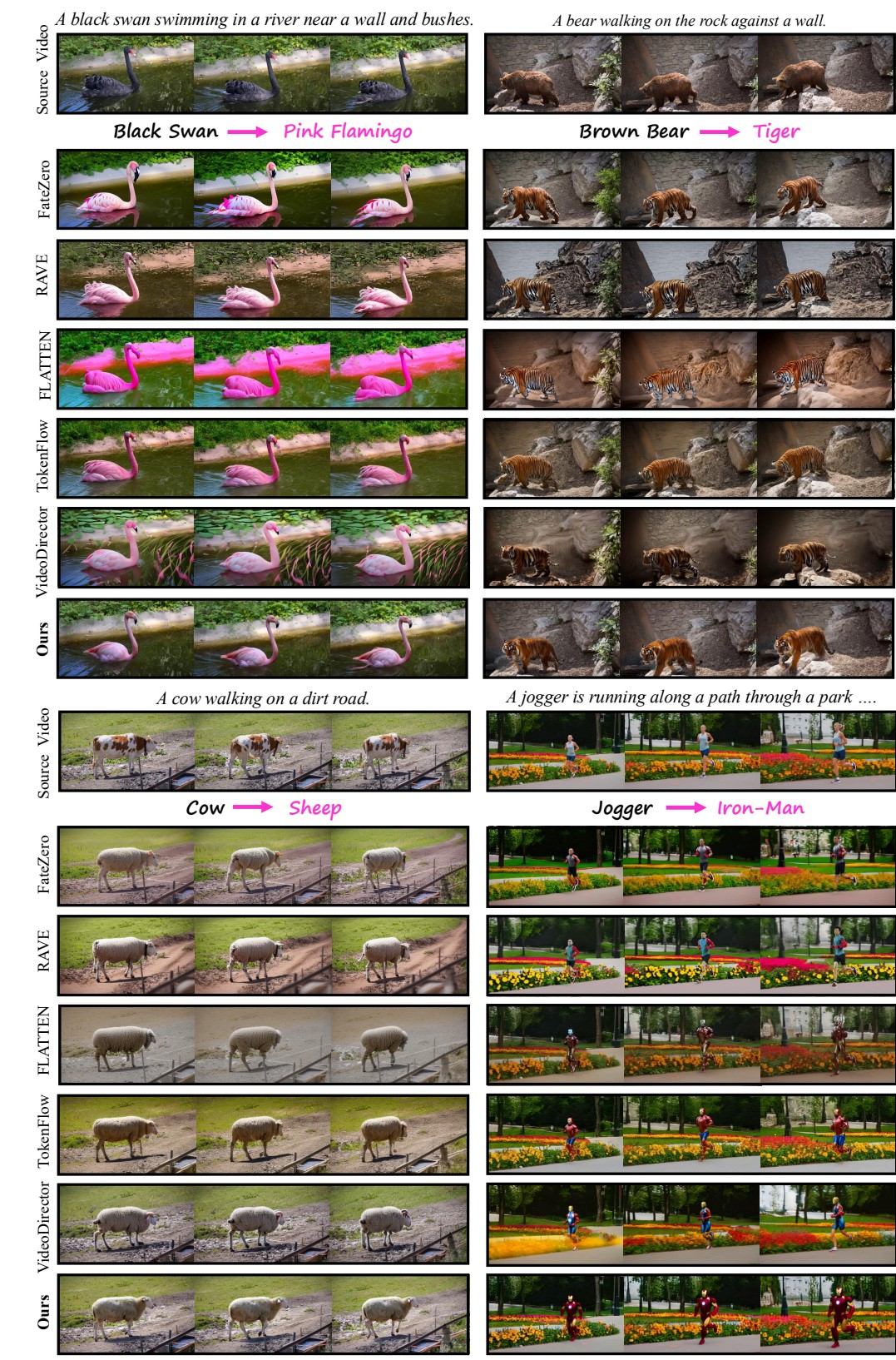

Figure 10: **More Visual Comparison Results**. We provide a more qualitative comparison with the existing method. VEFLOW demonstrates more stable, precise, and user-intention-aligned editing performance.

Figure 11: **More Visual Editing Examples**. We showcase more editing examples of our VEFLOW. Our approach provides a flexible video editing and achieves superior editing performance in cases such as subtle attribution change, small subject, etc.

