# OpenReview forum: "VEFlow: Training-free Text to Video Editing via Inversion-Free Video Editing Flow"
_ICLR.cc/2026/Conference — ICLR 2026 Conference Withdrawn Submission_

### Official Review · Reviewer_isHF · 2025-10-24

**Soundness:** 2
**Presentation:** 3
**Contribution:** 2
**Rating:** 4
**Confidence:** 4

**Summary:**

The paper introduces VEFlow, a training-free text-driven video editing method based on a spatio-temporal flow-guided diffusion pipeline.
It introduces a flow guidance module that propagates motion consistency between adjacent frames, ensuring temporal coherence without retraining. The approach leverages pre-trained image diffusion models and applies cross-frame latent warping guided by estimated optical flow.

**Strengths:**

1. Training-free, easy to integrate with existing diffusion models.
2. Maintains strong temporal consistency via flow propagation.
3. Solid experimental results on multiple editing tasks.
4. Clear and reproducible pipeline.

**Weaknesses:**

1. Limited novelty in algorithmic design.
The main contribution lies in integrating flow-based warping into an existing diffusion framework. While effective, the technique closely resembles prior works that use optical flow for temporal alignment or feature propagation (e.g., in ControlVideo, Rerender-A-Video, or TokenFlow).
The paper could better clarify what distinguishes its flow guidance from these approaches beyond being “training-free.”

2. Lack of robustness analysis.
VEFlow assumes reliable optical flow across frames, but no evaluation is provided for scenarios involving large motion, occlusion, or scene cuts. Since the method depends heavily on flow accuracy, it would be valuable to show how errors in flow estimation affect final quality.

3. Ablation study is incomplete.
The paper compares with baselines but lacks fine-grained ablations, e.g., removing or replacing the flow module, varying flow strength or consistency weights, testing different flow estimators.   These would strengthen the causal link between the proposed flow guidance and temporal improvement.

4. No discussion on computational overhead.
Flow estimation adds extra computation at inference time. The paper omits runtime or efficiency comparisons, which are relevant for “training-free” pipelines intended for practical deployment.

5. Limited quantitative evaluation.
The metrics used (FID, CLIP-score, LPIPS) do not directly measure temporal consistency. Adding metrics like tLPIPS or optical-flow-based temporal error would provide more convincing evidence of consistency gains.

**Questions:**

1. How sensitive is VEFlow to optical flow errors?
2. Could the model benefit from learned flow refinement or self-correction?
3. How does inference time compare to baselines like Text2Video-Zero?

---

### Official Review · Reviewer_y2Rj · 2025-10-29

**Soundness:** 2
**Presentation:** 3
**Contribution:** 2
**Rating:** 4
**Confidence:** 4

**Summary:**

Previous video editing approaches adopt an inverse-then-edit pipeline, but they suffer from unintended region editing and inefficiency. To address these limitations, the authors propose an inversion-free method. They further introduce attention-guided flow masking (AFM) and decoupled flow modulation (DFM) to strengthen performance. As a result, the proposed method demonstrates superior background preservation compared to prior work.

**Strengths:**

- **Clear and intuitive writing:** The paper is well-written and easy to follow. The motivation and advantages over prior work are clearly articulated.
- **Strong performance:** The proposed method exhibits clear improvements over existing approaches, both qualitatively and quantitatively.

**Weaknesses:**

- **Limited novelty:** The method appears to be primarily an extension of FlowEdit to video. AFM seems analogous to PTP-style blending, except applied to flow rather than latent space, and thus produces similar outcomes. In essence, DFM appears to be the only original contribution. Overall, this feels like FlowEdit adapted to video with only incremental novelty, which may not be sufficient. Moreover, these techniques do not appear specific to video; they could be considered general techniques. It would be preferable to highlight advantages specific to video editing.
- **Limited comparisons:** Although several models are compared, Video-PTP is not included. Despite relying on inversion, other baselines do as well; moreover, Video-PTP inherits latent-blending from PTP and is a strong baseline. It should be compared against. Ideally, the authors could apply Video-PTP to the same base model (Wan 2.1) to allow a fair comparison.

**Questions:**

- At line 137, $Z_0^{edit}$ and $Z_1^{edit}$ appear to be reversed relative to the notation in line 126.
- I understand that FlowEdit struggles when editing large regions such as backgrounds. Does the proposed method share this limitation? For example, what happens when the region to be modified is the background?
- Attention masking seems crucial for background preservation, but an ablation of mask computation is missing. Are the values used here empirically chosen?

---

### Official Review · Reviewer_aUTw · 2025-10-30

**Soundness:** 2
**Presentation:** 2
**Contribution:** 3
**Rating:** 4
**Confidence:** 3

**Summary:**

The  paper proposed a training-free and inverse-free editing method.  The paper construct the velocity field from source images to target images. Then propose the AFM and DFM method to further improve editing results. The results shows effectiveness of method while provide some ablation on proposed methods.

**Strengths:**

The authors propose estimating the velocity field between the distributions of source and target images to achieve an inverse-free editing approach. This is a novel and efficient strategy.
The method employs attention masks to edit target regions in a straightforward yet effective manner. Additionally, the proposed flow modulation technique helps adjust the velocity direction toward the target image distribution.
The approach demonstrates superior performance compared to existing methods in terms of both effectiveness and efficiency. The provided visual results are compelling and support the claims.

**Weaknesses:**

1: The method appears to overlook the fact that the text conditions for the source and target images may differ. The model is parameterized as vθ​(zt​,t), rather than vθ​(zt​,t,c), where c represents the text condition. Ignoring the text condition is problematic when the source and target prompts differ could result in the whole method is hard to follow.  Please carefully build formulas to make sure it is correct and matches with the facts.

2: Some notations and formulas are unclear or even incorrect. For instance, in lines 105–106, the statement “The generative flow models attempt to construct a transportation path between the distributions of two random vectors” is quite confusing, the flow matching model construct velocity field between data distribution and noise distribution, not "two random vectors". Additionally, equations such as Zt=(1−t)X0+tX1Zt​=(1−t)X0​+tX1​ are difficult to follow—why is Zt​ used instead of Xt? (Line 115)

**Questions:**

How exactly is "portion inversion" and the category labeled "others" in Figure 1 computed? Could you provide a specific explanation of what costs are included in "others"?

Are there any experiments conducted to evaluate the application of AFM and DFM at different timesteps? The appendix only mentions the range [5, 25]. Is there any analysis on how AFM and DFM perform individually, such as whether they are suitable for different timesteps or how they might influence each other?

---

### Official Review · Reviewer_rGgd · 2025-10-30

**Soundness:** 2
**Presentation:** 3
**Contribution:** 1
**Rating:** 0
**Confidence:** 3

**Summary:**

The paper proposes a T2V editing tool. Unlike typical inversion-based diffusion models, flow-based approaches work on a direct path from source to target, which speeds up the overall editing process significantly. There are two further contributions, an Attention-guided Flow Masking (AFM) and a Decoupled Flow Modulation (DFM). The first enables editing on a particular area of the video, while leaving the rest of the video untouched, which is basically utilized as a background preserving measure. DFM mitigates potential conflicts between the source and target space. An extensive number of examples are shown to demonstrate the quality of results. An ablation study demonstrates the effectiveness of AFM and DFM, e.g., background preservation is quantitatively shown using a variety of distance metrics to the original background.

**Strengths:**

A direct data space-based T2V editing tool is certainly useful, and the editing efficiency is demonstrated. The proposed tool is more than twice as fast as the best competitor, which is a crucial characteristic for a manual editing tool. The derived results look convincing, and the ability to alter certain parts of the video without impacting other parts seems to be very valuable. Apart from some minor mistakes, the paper is well written. In addition to many examples, a couple of experiments provide additional insights to the reader, as the deformation of masks and the effect of DFM with increasing w. An extensive appendix provides further results,  explanations, and further experiments and ablation studies.

**Weaknesses:**

There are serious doubts about the novelty of the paper and the completeness of the state of the art. There is a publication on arxiv and corresponding code on github from a method called FlowDirector (https://flowdirector-edit.github.io/, https://arxiv.org/pdf/2506.05046) by Guangzhao Li et. al. The ICRL26 paper deadline was on Sep 24 '25, and the paper and the code of FlowDirector were presented on Jun 5 '25, which is a close to 4-month gap. FlowDirector is very similar to the proposed approach. There is an inversion-free approach with Spatially Attentive Flow Correction (SAFC), and this SAFC seems to be the same as AFM. The black swan to white duck example (Fig. 3), which seems to be the most outstanding result in the proposed paper, is part of Flow Director as well, with nearly identical results. As my task as reviewer, how can I distinguish whether the results in the proposed paper are genuine or whether the code of Guangzhao Li et. al. was used to generate a few examples?  In any case, a comparison of VEFlow with FlowDirector should be mandatory. Taking a look at the first claim under contributions: "We introduce a novel flow-based inversion-free video editing framework, which eliminates the explicit inversion process, offering a new perspective on text-driven video editing." This can be understood as a claim to be the first one doing this, which is obviously not the case, since in FlowDirector it is read "We propose FlowDirector, a novel inversion-free video editing framework.". On a brief inventory, it does not look as if VEFLow provides better results than FlowDirector, and the background preservation scores on SSIM seem to be similar as well.

In Fig. 4, Baseline+AFM+DFM, you can see a green meadow between the robot's legs, which was obviously generated by the model. Unfortunately, there is no analysis of failure cases and no discussion of limitations. But this clearly shows a disadvantage of a mask-based approach. The green tone of the inserted meadow does not match the color distribution of the background. I assume that this will occur very frequently, even though the paper apparently aims at showing positive examples only. The same problem occurs in the examples rabbit to tiger (green tone between the legs) and with all examples in Fig. 7 in the appendix.

In Fig. 3, the comparison methods manage to transfer the pose of the Jeep and the bear to the Porsche and the lion, respectively. More precisely, the Jeep in the right-hand image is driving from right to left. The bear tends to look away from the camera toward the wall in the background. In the proposed method, the Porsche drives from back to front, and the lion's head faces the camera. The comparison methods do a better job of this aspect. This problem obviously also occurs in the woman-to-Spider-Man example in Fig. 11. In Fig. 3, the illumination of the lion does not match the background. This is a similar problem to that of the lawn in example Fig. 4, where the color distribution is not correct. It is correct that the other methods alter the background more than the proposed method, but they create a more harmonious overall picture. This seems to be the trade-off of the process: background preservation at the expense of image realism. Thus, I do not share the view that the proposed method outperforms the other methods, and in terms of editing alignment, it seems to fall short in this comparison, since the pose is not preserved.

I strongly advise the authors of the paper to replace one example in Fig. 11. First, the prompt obviously did not work since the man in the result image is wearing pants and is fortunately not naked. In addition, the prompt from Man-to-Naked Man is likely ethically problematic and illegal in some countries around the world. There is a major ethical debate about how to fight and limit deepfake pornography.

Minor errors:
- In Fig. 7 there is a reference to Eq. 7, yet this is probably meant to be Eq. 6.
- In line 121, is should probably be "Instead of taking the inversion-based approach"

**Questions:**

The results in Table 2 are unexpected. While it is clear that AFM plays a crucial role in background preservation, the comparison to the baseline perfectly demonstrates this. Yet, why is the background even better preserved when using DFM in addition? In terms of PSNR, the effect of DFM seems to be as large as the effect of AFM. This is surprising because in nearly all examples, the foreground region seems to be only 10 to 30 percent of the frame size. Thus, the mask affects a large area of 70 percent or more. DFM seems to operate mainly on the foreground region, like the overlapping region of the jogger and the Iron Man in Fig. 10. From this small region, most is not labeled background at all. The effective area that is measured is probably very small. How can such a small area have such a huge impact on the result?  The improvements in this area must be enormous. This is an absolutely stunning result, which should definitely be investigated in more detail, and these results could give valuable insights to the reader. Furthermore, the scores PSNR and MSE are closely linked to each other: $PSNR = 10 \cdot log_{10} (\frac{Max^2}{MSE})$. Surprisingly, Baseline+AFM is by far better in terms of MSE, while Baseline+AFM+DFM is by far better in terms of PSNR. How is this possible?

**Details Of Ethics Concerns:**

One example in Fig. 11 (appendix) provides an editing prompt: "Man to Naked Man". This may be ethically problematic and could be illegal in many countries around the world. There is a major debate on how to fight and limit deepfake pornography, especially child pornography. I doubt that ICLR wants to provide a platform and advised the authors to remove this particular example.

---

> ### Author Response · Authors · 2025-11-14
>
> We thank the reviewer for their time and insightful comments. We have checked all the points raised and will revise the manuscript accordingly. Below, we provide clarifications to prevent potential misunderstandings of our work.
>
> - *Clarification with Concurrent Work*: We thank the reviewer for pointing out the concurrent work, FlowDirector (FD), which we sincerely apologize for missing in our initial submission. We would like to clarify the key distinctions between our work and FD: (1) While both works explore a similar idea, we **did not agree** with the reviewer's comment that our AFM module was the same as the SAFC proposed in FD; instead, our AFM exhibits a different design philosophy from SAFC. Specifically, FD employs a complex process involving spatial smoothing and edge softening (e.g., ComputeDistanceTransform) to generate the mask from the cross-attention map. By contrast, we resort to simpler Morphological operations to obtain the complete mask, maintaining the overall pipeline light and efficient. These technical differences can be validated with the direct comparison of their paper and our draft. Furthermore, our additional DFM module also delivers a distinct design motivation entirely compared with FD, highlighting our unique contribution. (2) We argue that it is **irresponsible** to claim that our results of the black swan-to-white duck example (Fig. 3) were possibly obtained by running FD's code **without any evidence.**  In fact, we developed our method and codebase independently. Moreover, the swan-to-white duck editing example is a classic and widely used common illustrative case in video editing. As a reviewer, it is improper to **make a malicious speculation emotionally** that our results may be generated by FD's code, just because some similar results are also presented in FD's paper. We will also open-source our code upon acceptance to facilitate the transparency check. (3) We appreciate the reviewer's feedback, and we are committed to explicitly discussing FD and making the direct comparison with it in our revised manuscript.
>
> - *Clarification on the Results*: We sincerely thank the reviewer for taking the time to examine our results in such detail. (1) *On the Subtle Blending Defect*: Firstly, we would like to emphasize that it does not occur frequently in our practice, and the reviewer made a very **subjective judgment without any evidence.** There may be some occasional blending artifact between the edited foreground and background in some cases. However, it is subtle and is not a frequent failure mode in our experiments. In fact, we have conducted analysis (detailed in Fig. 7 of the Appendix) and identified a mitigation strategy. Specifically, we found that this issue can be significantly alleviated by adjusting the timesteps of our masking strategy. Specifically, by reserving a portion of the later denoising steps to jointly refine the high-frequency details at the foreground-background boundary, we can produce more natural and harmonious results without compromising edit quality. (2) *On the Pose Consistency:* In the context of text-driven video editing, we argue that the objective is not pixel-level pose transfer, but rather the preservation of the **overall motion dynamics** of the original video. The text-based editing instruction inherently grants the model a certain degree of freedom to adapt the subject's pose and movement to better suit its new appearance (e.g., a "lion" turning its head differently than a "bear" might in the same original motion sequence). We believe this adaptive behavior is a strength that leads to more natural and physically plausible outputs, as the motion can organically align with the new subject's characteristics. We acknowledge that our method is not perfect. However, we believe our approach represents a more advanced video editing method compared with existing methods in terms of the edit results and efficiency.  We also thank the reviewer for their constructive comment. We will add the limitation discussion in our revised version.
>
> - *Results in Table. 2*: We greatly appreciate the reviewer's meticulous attention to detail in identifying the inconsistency in Table 2. Upon a thorough investigation, we discovered some bugs during our evaluation of the AFM module ablation study, in which the wrong results video paths are passed during evaluation, causing different sets of videos to be used for calculating MSE versus the other metrics (LPIPS, PSNR, etc) and leading to incorrect evaluation results. We have since fixed these bugs, re-run the entire evaluation for the AFM ablation with the correct and consistent videos, and updated Table 2 accordingly.
>
> - *Ethical Concern*: We sincerely apologize for this oversight. This example will be removed entirely and replaced with a more appropriate one in the revised version of the paper. We are carefully reviewing all our examples to ensure they adhere to the ethical standards for publication.

---

### Note · Authors · 2025-11-14

I have read and agree with the venue's withdrawal policy on behalf of myself and my co-authors.